# A Non-Equal Time Interval Incremental Motion Prediction Method for Maritime Autonomous Surface Ships

**DOI:** 10.3390/s23052852

**Published:** 2023-03-06

**Authors:** Zhijie Zhou, Haixiang Xu, Hui Feng, Wenjuan Li

**Affiliations:** 1Key Laboratory of High-Performance Ship Technology, Wuhan University of Technology, Ministry of Education, Wuhan 430063, China; 2School of Naval Architecture, Ocean and Energy Power Engineering, Wuhan University of Technology, Wuhan 430063, China; 3Marine Equipment Technology Institute, Jiangsu University of Science and Technology, Zhenjiang 212003, China

**Keywords:** maritime autonomous surface ship, cubature Kalman filter, incremental motion prediction, long short-term memory

## Abstract

Recent technological advancements facilitate the autonomous navigation of maritime surface ships. The accurate data given by a range of various sensors serve as the primary assurance of a voyage’s safety. Nevertheless, as sensors have various sample rates, they cannot obtain information at the same time. Fusion decreases the accuracy and reliability of perceptual data if different sensor sample rates are not taken into account. Hence, it is helpful to increase the quality of the fusion information to precisely anticipate the motion status of ships at the sampling time of each sensor. This paper proposes a non-equal time interval incremental prediction method. In this method, the high dimensionality of the estimated state and nonlinearity of the kinematic equation are taken into consideration. First, the cubature Kalman filter is employed to estimate a ship’s motion at equal intervals based on the ship’s kinematic equation. Next, a ship motion state predictor based on a long short-term memory network structure is created, using the increment and time interval of the historical estimation sequence as the network input and the increment of the motion state at the projected time as the network output. The suggested technique can lessen the effect of the speed difference between the test set and the training set on the prediction accuracy compared with the traditional long short-term memory prediction method. Finally, comparison experiments are carried out to validate the precision and effectiveness of the proposed approach. The experimental results show that the root-mean-square error coefficient of the prediction error is decreased on average by roughly 78% for various modes and speeds when compared with the conventional non-incremental long short-term memory prediction approach. Additionally, the proposed prediction technology and the traditional approach have virtually the same algorithm times, which may fulfill the real engineering requirements.

## 1. Introduction

In the global economy, marine transportation is becoming increasingly significant. More than two thirds of all freights in international trade are transported by sea. Therefore, the ability of marine transportation to link the globe and support technical advancement in maritime equipment is extremely important. In recent years, progressive intelligence based on digitalization and aiming for autonomy has emerged as a new trend and hot spot in the development of the shipbuilding industry, driven by the development of cutting-edge theories and technologies such as the Internet of Things, big data, and artificial intelligence (AI). The world’s major shipbuilding and shipping nations have expanded their investments in the creation and use of autonomous ships. Autonomous ships, also known as maritime autonomous surface ships (MASS), have been developed and are increasingly being used [1]. The desire to avoid human error—which contributes significantly to the bulk of maritime catastrophes—led to the creation of MASSs. Additionally, crewed ships have also been linked to high operating costs. Therefore, the requirement to avoid the financial expenditures and human mistakes linked to crewed ships serves as the main driving force behind developments in autonomous shipping [2].

The autonomous degree of MASS has been divided into L1–L4 in the 100th Maritime Safety Committee (MSC) by the International Maritime Organization (IMO) [3]. The mode of the perception module in a ship’s autonomous navigation system changes from a shared view of mariners and machines into a total machine perception as the degree of autonomy develops, which results in an increase in the number and variety of sensors carried by MASS. Therefore, research into multi-sensor fusion technology in autonomous navigation systems is crucial to raising the level of ship autonomy.

The perception data of a ship’s autonomous navigation system are mostly composed of information about the ship’s motion state (such as its position, heading, speed, attitude, etc.), as well as information about the surrounding environment (such as the status of other ships, pontoons, and other obstacles that may threaten the navigation safety of the ships). Nevertheless, the accuracy of the former’s perception serves as the primary guarantee of the latter’s perception correctness. For instance, if the perception of a ship’s own motion status is erroneous, even though the relative azimuth information of an obstruction perceived by the shipborne sensor has adequate precision, the absolute coordinates of the obstacle will be guessed incorrectly.

The perception of a ship’s own motion status is mostly based on the fusion information provided by shipboard positioning systems and inertial navigation systems (INSs), such as the global positioning system (GPS), global navigation satellite system (GLONASS), Galileo positioning system, BeiDou, a compass, and a gyroscope. In the past decade, researchers have carried out various research works on developing efficient multi-sensor fusion algorithms to estimate the motion status of ships. For instance, a fusion technique based on the Kalman filter (KF) and particle filter (PF) was developed [4]. This method accurately estimated the status of a ship by fusing its position and attitude data. To further improve the accuracy of ship motion attitude estimates, Ref. [5] proposed a novel transfer alignment method for a gimbaled inertial navigation system (GINS) and strap-down inertial navigation system (SINS) based on an iterative computation approach. By using this technique, the alignment complexity is reduced. Additionally, a dynamic positioning ship state estimation technique combining the unscented Kalman filter (UKF) and PF was also presented in [6]. In this approach, the UKF optimizes each particle state update while the PF serves as the general framework. Then, using the particles’ importance distribution, the low-frequency condition of a ship’s motion was determined. Ref. [7] focused on the least-squares linear fusion filter design for discrete-time stochastic signals from a multi-sensor. A covariance-based approach was used to derive easily implementable recursive filtering algorithms under centralized, distributed, and sequential fusion architectures. The year after, using the linear minimum variance optimality criterion, the local least-squares linear filter obtained with each sensor was improved as a distributed fusion filter with a matrix-weighted linear combination while taking into account the autocorrelation and cross-correlation of multi-sensor measurements [8]. Furthermore, considering the random packet loss in the transmission process of sensor measurement signals, a recursive filtering algorithm was designed by using a method based on covariance and two compensation strategies based on measurement prediction [9]. In this algorithm, an alternative method based on direct estimation measurement noise and innovative technology was used to compensate for the packet loss system. To make the navigation system more reliable, Ref. [10] proposed an improved RISS-GPS ship navigation approach. Modern magnetometer and azimuth calibration technology served as the foundation for this technique.

The research on the fusion technology of a ship’s own motion state has gradually matured and is widely used. The following are its main steps: The mathematical model of a ship’s motion is first discretized, and then the measurements from each sensor are used as the input to perform the fusion estimation of the step with an equal time interval based on the framework of the Kalman filter and its enhanced method. Nevertheless, sampling frequency varies widely since there are so many different types of sensors on board. The sampling rate of ambient sensors, such as cameras and lidar, is typically between 10 and 30 Hz, whereas the sampling rate of GPS is typically between 1 and 5 Hz. As a result, the latest estimate of a ship’s state of motion at the time ambient sensors take samples is often tens or even hundreds of milliseconds prior to the sampling time. The higher the speed of a ship, the more significant the difference in motion state will be. As a consequence, matching the sample from an external environment detecting sensor with the outcomes of a ship motion state estimate in the time dimension is the key technology in fusion technology.

Currently, ship motion status prediction makes considerable use of the long short-term memory (LSTM) model. It may learn the features of ship motion through historical observations, and then predict the motion state at a specific future time. However, this method is often used to predict a ship’s trajectory with strong regularity and is rarely used to predict a ship’s irregular motion directly, such as roll and pitch motion. The method, however, does not account for the modification of the motion law brought on by varying speeds. This paper proposes a non-equal time interval incremental prediction (NETIIP) method. First, a collection of cruise data is trained offline using the LSTM architecture. The time deviation and state increment are used as inputs, and the increment of actual state at the moment to be estimated is taken as the output of the network. Then, using the cubature Kalman filter (CKF) estimator, the position, attitude, speed, and other characteristics of the ships are projected at regular intervals. The estimation result of the ship motion state at this moment is finally derived based on the network estimation from offline training after receiving the time stamp of the environmental measurement sensor. The advantages of this approach are as follows:(a)NETIIP uses the CKF estimates as the input for the LSTM prediction network rather than the sensor’s original data. On the one hand, it can effectively avoid the problem of reduced prediction accuracy caused by sensor measurement error signals as the object of network learning. On the other hand, the amplification of sensor measurement noise caused by first-order state difference can be suppressed.(b)NETIIP adopts a semi-supervised learning mode, which not only learns the changes in a ship’s position and attitude but also incorporates the changes in the ship’s speed into the learning features of the network, which can minimize the impact of the poor learning performance caused by ship speed differences. It merely needs to learn annotated datasets of changes in ship movement at any speed. It is feasible to foresee the ship’s motion status at various speeds.(c)NETIIP employs the technique of learning the properties of state increments rather than directly learning the features of motion states. As opposed to the non-incremental LSTM prediction (NI-LSTM) method, it avoids the shortage of the poor network learning rate caused by the difference between the state characteristics of various speeds or sailing modes and the state characteristics of the training set.

The rest of this paper is organized as follows. The related works on the key technologies for matching the information in an asynchronous system are introduced in Section 2. In Section 3, the NETIIP process and algorithm modules are provided. Section 4 contrasts the NETIIP approach and the NI-LSTM method using contrast model experiments, and the NETTIP method is proved to be effective and feasible. Some conclusions are given in Section 5.

## 2. Related Works

### 2.1. Establishing Fusion Model

Establishing an asynchronous fusion model is one of the key technologies for solving the problem of information matching in asynchronous systems. Some scholars have conducted related research on such techniques. A cubature Kalman interactive multi-modal fusion algorithm was one of these methods [11]. After initially generating the models under different frequencies based on the observed values of different sampling frequencies, this algorithm used the higher frequency fusion period, and the interactive multiple model (IMM) approach was used to weight the predicted values of each model. Furthermore, Ref. [12] proposed two distributed asynchronous fusion algorithms, batch generalization covariance intersection (B-GCI) and sequential generalization covariance intersection (S-GCI), for the asynchronous sensor fusion problem for an arbitrary number of sensors with different sampling rates within the framework of the random finite set (RFS) theory. S-GCI was utilized to lessen the B-heavy GCI’s computational load, while B-GCI was used to avoid the challenging computations of cross-covariance among the local posteriors of sensors. To provide an asynchronous track fusion technique with information feedback, an asynchronous track fusion model with an irregular time interval of the observation data and in conjunction with the track quality with multiple model (TQMM) was developed in [13]. The asynchronous multi-sensor fusion system’s performance was improved by the method’s weight allocation, which made use of the TQMM. The unique issue of each sub-navigation sensor in the multi-sensor integrated navigation system with data sampling rates that were different and several times the rational rate was addressed by reconstructing the state equation and measurement equation of the system and basing it on the matrix operator with scale and wavelet properties [14]. Moreover, Ref. [15] studied the multi-target tracking problem based on an asynchronous network of sensors with different sampling rates. This method carried out an arithmetic averaging approach, recursively, in a timely manner according to the network-wide sampling time sequence in order to fuse the filter estimates obtained at different sensors conditioned on asynchronous measurements.

### 2.2. Time Alignment

Time alignment is another strategy for overcoming asynchronous fusion. Local estimation results for varied period sizes are brought into agreement with the uniform time by employing timestamp alignment. To ensure that the time alignment parameters converged well, a convex combination of the instantaneous cost and the squared difference between current and past estimates was proposed, and arithmetic average fusion was used to merge the intensities from various nodes [16]. Additionally, Ref. [17] proposed batch time-aligned asynchronous fusion using unbiased converted measurements. The connection between measurement error covariance and the actual measurement was used to demonstrate that this approach is theoretically suboptimal. Ref. [18] believed that applying the time alignment would result in additional estimation error. In order to address this issue, a technique for asynchronous track-to-track association without time alignment was presented. In this algorithm, the deviation in the time sequence was considered as the deviation in each local estimated track.

As can be seen from the research mentioned above, three categories can be made for asynchronous multi-sensor fusion techniques. The first is creating an asynchronous fusion model and feeding it inputs that are sensor measurements at varied sampling rates. Theoretically, this method is the most accurate, but creating a workable model for asynchronous fusion is difficult. Inaccurate or imprecise models will result in a reduction in fusion accuracy. The second method is distributed timing alignment. The time stamp alignment method is used to align the results of each local estimation after sensors with various sampling frequencies are estimated locally. This method is simpler to implement than the first, but the key to ensuring the correctness of the fusion using this method is to figure out how to eliminate the excess mistake. The final approach disregards the timing problem and considers it as a measurement error for direct fusion. This method performs better with measuring systems that change slowly, but it has the lowest accuracy and the least amount of theoretical support when the thing being measured changes fast and often.

The research shows that it is difficult to create an asynchronous sampling model based on the self-positioning sensor group and the ambient sensing sensor group, which are not independent of one another, for the navigation systems of MASS. In other words, the information gathered through environmental sensing is reliant on a ship’s own attitude and position. Therefore, the timing alignment method is more suitable for this paper. Using the method of prediction to achieve time alignment can not only consider the change law of the system with time but also has no need to establish an accurate model, which is easy to realize in a practical engineering application.

## 3. Materials and Methods

### 3.1. Problem Statement

The sample frequency of each sensor is not exactly the same in a heterogeneous multi-sensor system. For instance, the positioning sensor’s sample frequency in a MASS intelligent navigation system is typically 1 to 5 Hz (such as GPS), whereas a visible light camera and other sensors that are frequently used to collect data from the environment have a sampling frequency that is typically greater than 20 Hz. Figure 1 depicts the sampling diagram for a kind of heterogeneous multi-sensor.

In Figure 1, $\Delta t_{a}$ and $\Delta t_{b}$ represent the length of the time difference between the continuous sampling times for camera.

Because the estimating technique often requires much less time than a positioning sensor’s sampling period, the equal time interval estimation strategy is frequently employed to estimate a ship’s own motion state. Nevertheless, the complexity of the external environment for a sensor of the external environment is positively connected to the time of the recognition and localization algorithm of the external target. The length of time required for detection and localization algorithms varies depending on how complicated the environment is (as illustrated in Figure 1, $\Delta t_{a}\neq \Delta t_{b}$). In other words, the sample interval used by an external environment sensing sensor can be thought of as not being equal.

In a ship’s autonomous navigation system, the sensing module must simultaneously superimpose the information gathered by the external environment sensing sensor and the state vector of the sensor’s measurement coordinates origin in order to determine the type of the external target obstacle and its absolute position coordinates in real-time. The origin state of sensor measurement coordinates is related to the position and attitude of the ship, etc. If the offset and rotation of the sensor installation position are not taken into account, and the measurement coordinate system of the sensor coincides with the coordinate system of ship movement, the relationship can be described as follows:(1)$$\eta_{absolute}=\eta_{ship}+\eta_{relative}$$
where $\eta_{absolute}$ refers to the motion state of the external target obstacle in the geodetic coordinate system (absolute state), $\eta_{ship}$ is the present motion state of the ship, and $\eta_{relative}$ is the motion state of the target measured with the sensor (relative state). However, because of the difference in frequency, Formula (1) is described as (taking the estimation at time $t_{2}$ as an example):(2)$$\eta_{absolute}(t_{2})=\eta_{ship}(t_{2})+\eta_{relative}(t_{2})\neq \eta_{ship}(t_{1})+\eta_{relative}(t_{2})$$
where $\eta_{\cdot }(t)$ means the motion state at time t.

Therefore, one need for guaranteeing the sensing accuracy of a ship’s autonomous navigation sensing module is to anticipate the ship’s motion state at uneven time intervals according to the value $\eta_{ship}(t_{1})$ at equal time intervals.

### 3.2. Design of the Prediction Process

To ensure the real-time performance of the heterogeneous multi-sensor system and reduce the computation required for the prediction process, a marginal computing framework is employed, as shown in the flowchart in Figure 2.

In Figure 2, $\eta_{\cdot }$ represents the measurement value of the sensor, while ${\hat{\eta }}_{\cdot }$ represents the estimated value of the state.

The NETIIP algorithm is mostly composed of two components. The first section is the equal time interval estimation of a ship’s own motion state, in which the CKF estimator is used to carry out a real-time estimation of the ship’s 6-DOF motion state in accordance with the position and navigation sensor data sampled at the same period. The results ${\hat{\eta }}_{ship}(t_{1})$ are stored in the memory register.

The second component is a state predictor built using the LSTM framework. First, according to the time stamp sampled at irregular intervals, the most recent historical moment dataset is taken from the memory register. The ship motion state at time $t_{2}$ is then predicted using the LSTM network based on offline training, and the network is simultaneously updated. In contrast to the conventional NI-LSTM network, the network will additionally use the timestamp of non-equal interval sampling as an input variable to realize non-equal interval state prediction. In the subsections that follow, we go into further detail on the designs of the two pieces.

### 3.3. Ship Motion Mathematical Model

The kinematic model and dynamic model are two categories for the mathematical representation of ship motion [19]. The dynamic model is primarily used to examine the link between a ship’s propulsion and external environmental factors on the ship’s motion, whereas the kinematic model treats the ship as a particle and is primarily used to study the change in the ship’s position and heading over time. In contrast to the kinematic model, the dynamic model is built using the ship’s precise hydrodynamic characteristics, thrust force, and environmental force. Accurately obtaining these variables for the application is very challenging. As a result, we estimate the ship’s 6-DOF motion state using the ship kinematic model.

Firstly, as shown in Figure 3, the geodetic and shipboard coordinate systems are constructed.

All coordinate systems in Figure 3 follow the right-handed criterion, where O−NED is the geodetic coordinate system, also known as the navigation coordinate system, while $O_{B}-X_{B}Y_{B}Z_{B}$ is the shipboard coordinate system, with the center of gravity of the ship as the origin, and the positive direction of $X_{B}Y_{B}Z_{B}$ points to the bow, starboard, and bottom of the ship, respectively. The ship moves in three axes of surge, sway, and heave as well as in three axes of roll, pitch, and yaw.

As a result, the 6-DOF motion status of the ship can be expressed as follows:(3)$$\eta_{ship}={\left[\begin{array}{ccc} x & y & \begin{array}{ccc} z & \theta & \begin{array}{cc} \varphi & \psi \end{array} \end{array} \end{array}\right]}^{T}$$
(4)$$\nu_{ship}={\left[\begin{array}{ccc} u & v & \begin{array}{ccc} w & p & \begin{array}{cc} q & r \end{array} \end{array} \end{array}\right]}^{T}$$
where $\Omega =\left[\begin{array}{ccc} x & y & z \end{array}\right]$ indicates the coordinate of $O_{B}$ in the O−NED and $\Theta =\left[\begin{array}{ccc} \theta & \varphi & \psi \end{array}\right]$ represents the ship’s roll angle, pitch angle, and heading at the current moment, respectively, and $\nu_{ship}$ represents the speed of the ship’s 6-DOF motion in $O_{B}-X_{B}Y_{B}Z_{B}$.

In Figure 3, according to the spatial coordinate transformation relation, the first derivative ${\dot{\eta }}_{ship}$ of the $\eta_{ship}$ has the following relation to $\nu_{ship}$:(5)$${\dot{\eta }}_{ship}=\left[\begin{array}{cc} R_{\Theta } & 0_{3\times 3}\\ 0_{3\times 3} & I_{3\times 3} \end{array}\right]\nu_{ship}=J_{\Theta }\nu_{ship}$$
where $I_{3\times 3}$ is a third-order identity matrix, $0_{3\times 3}$ is a third-order zero matrix, and $R_{\Theta }$ is the rotation matrix, which can be expressed as
(6)$$R_{\Theta }=R_{Z_{B}}R_{Y_{B}}R_{X_{B}}=\left[\begin{array}{ccc} c\psi & -s\psi & 0\\ s\psi & c\psi & 0\\ 0 & 0 & 1 \end{array}\right]\left[\begin{array}{ccc} c\varphi & 0 & s\varphi \\ 0 & 1 & 0\\ -s\varphi & 0 & c\varphi \end{array}\right]\left[\begin{array}{ccc} 1 & 0 & 0\\ 0 & c\theta & -s\theta \\ 0 & s\theta & c\theta \end{array}\right]$$
with *c* and *s* standing for cos() and sin() trigonometric operators, respectively.

When the discrete time is short enough, the 6-DOF velocity of the ship can be regarded as constant. Therefore, the ship’s motion can be described discretely as:(7)$$\eta_{ship}\left(t_{k}\right)=I_{6\times 6}\times \eta_{ship}\left(t_{k-1}\right)+{T\times \dot{\eta }}_{ship}\left(t_{k-1}\right)$$
where $T={diag}_{6\times 6}({\Delta t}_{GPS})$ represents the discrete time matrix.

Considering that the increment $\eta_{ship}$ is related to $\nu_{ship}$, it is necessary to estimate the state of $\eta_{ship}$ and $\nu_{ship}$ at equal time intervals. Therefore, the state ${\hat{Χ}}_{ship}$ to be estimated is
(8)$${\hat{Χ}}_{ship}={\left[\begin{array}{cc} {\eta_{ship}}^{T} & {\nu_{ship}}^{T} \end{array}\right]}^{T}$$

According to Equations (5)–(8), ship motion can be described as the following state–space equation:(9)$${\hat{Χ}}_{ship}\left(t_{k}\right)={\left[\begin{array}{cc} I_{6\times 6} & J_{\Theta }(t_{k-1})\\ 0_{3\times 3} & I_{6\times 6} \end{array}\right]}^{-1}\left[\begin{array}{cc} I_{6\times 6} & T\\ 0_{3\times 3} & I_{6\times 6} \end{array}\right]\left[\begin{array}{cc} I_{6\times 6} & J_{\Theta }(t_{k-1})\\ 0_{3\times 3} & I_{6\times 6} \end{array}\right]{\hat{Χ}}_{ship}\left(t_{k-1}\right)$$
where $J_{\Theta }(t_{k-1})$ and ${\hat{Χ}}_{ship}\left(t_{k-1}\right)$ are all related to $\Theta \left(t_{k-1}\right)$. Therefore, the ship motion is nonlinear.

However, considering that the common shipborne sensors cannot observe all the states in the ${\hat{Χ}}_{ship}$, we only use the sensor group (e.g., GPS and INS) to observe $\eta_{ship}\left(t_{k}\right)$:(10)$$Z_{ship}^{1}\left(t_{k}\right)={\left[\begin{array}{ccc} x^{GPS}\left(t_{k}\right) & y^{GPS}\left(t_{k}\right) & \begin{array}{ccc} z^{GPS}\left(t_{k}\right) & \theta^{INS}\left(t_{k}\right) & \begin{array}{cc} \varphi^{INS}\left(t_{k}\right) & \psi^{INS} \end{array} \end{array} \end{array}\left(t_{k}\right)\right]}^{T}$$
where $\cdot^{GPS}$ and $\cdot^{INS}$ indicate that the state is observed using GPS and INS, respectively.

Due to the limited accuracy of GPS for the ship’s current elevation measurement, which may reduce the accuracy of estimation, $z^{GPS}$ are set to 0. Moreover, the time difference method is used to observe the velocity state $\nu_{ship}\left(t_{k}\right)$:(11)$$Z_{ship}^{2}\left(t_{k}\right)={J_{\Theta }\left(t_{k}\right)}^{-1}\left(\frac{Z_{ship}^{1}\left(t_{k}\right)-Z_{ship}^{1}\left(t_{k-1}\right)}{\Delta t_{GPS}}\right)$$

The observation corresponding to Equation (9) of state can be expressed as:(12)$$Z_{ship}\left(t_{k}\right)={\left[\begin{array}{cc} {Z_{ship}^{1}\left(t_{k}\right)}^{T} & {Z_{ship}^{2}\left(t_{k}\right)}^{T} \end{array}\right]}^{T}$$

Equations (9)–(12) demonstrate that the equations for state and observation are both nonlinear; hence, the nonlinear estimating method must be used to estimate the motion state of the ships over equal time intervals.

### 3.4. Equal Time Interval Estimation Method for Discrete Nonlinear Systems

Conventional linear system estimating methods such as the Kalman filter cannot be used to estimate ship motion in nonlinear systems in real-time. As a result, nonlinear estimation techniques, such as the extended Kalman filter (EKF) [20], the UKF [21], and the CKF [22], are frequently utilized for the estimation of ship motion. As nonlinear filtering techniques, both the EKF and UKF can accomplish real-time state estimation in nonlinear ship motion systems. Meanwhile, the UKF offers the advantages of quicker convergence, higher filtering precision, and easier implementation [23]. However, when estimating in nonlinear systems with high dimensions (dimensions greater than three), the CKF performs more accurately than the UKF [24]. Next, we will introduce the calculation flow of the two popular nonlinear system point estimate methods, the CKF and UKF, and the flow charts are shown in Figure 4.

#### 3.4.1. CKF Estimation Method

The CKF is based on the third-order spherical–radial cubature criterion and uses a set of cubature points to approximate the mean and covariance of states for nonlinear systems with additional Gaussian noise, which is theoretically the closest approximation algorithm to Bayesian filtering. The CKF mainly consists of two parts—time update and measurement update—and the algorithm flow is as follows.

(I)Time Update

Step 1: Perform Cholesky decomposition for the error covariance $P_{t_{k-1}}$ at time $t_{k-1}$:



(13)
$$P_{t_{k-1}}=S_{t_{k-1}}{S_{t_{k-1}}}^{T}$$



Step 2: Select cubature point (*i* = 1, 2…, 2*n*):

(14)$$\chi_{i,t_{k-1}}=S_{t_{k-1}}\xi_{i}+{\hat{Χ}}_{ship}\left(t_{k-1}\right)$$(15)$$\xi_{i}=\sqrt{n}{\left[I_{n}-I_{n}\right]}_{i}$$
where *n* = 12 represents the dimension of ${\hat{Χ}}_{ship}\left(t_{k-1}\right)$,$I_{n}$ is the identity matrix of order *n*, and ${\left[I_{n}-I_{n}\right]}_{i}$ is the *i*th column vector of the matrix $\left[I_{n}-I_{n}\right]$.

Step 3: Propagate the cubature point through Equation (9):



(16)
$$\chi_{i,t_{k-1}}^{⋆}=f(\chi_{i,t_{k-1}})$$



Step 4: Estimate the predicted value of state at time $t_{k}$:



(17)
$${Χ}_{ship}^{⋆}\left(t_{k}\right)=\frac{1}{2n}\sum_{i=1}^{2n}\chi_{i,t_{k-1}}^{⋆}$$



Step 5: Estimate the predicted value of the error covariance at time $t_{k}$:

(18)$$P_{t_{k}}^{⋆}=\frac{1}{2n}\sum_{i=1}^{2n}\chi_{i,t_{k-1}}^{⋆}{\chi_{i,t_{k-1}}^{⋆}}^{T}-{Χ}_{ship}^{⋆}\left(t_{k}\right){{Χ}_{ship}^{⋆}\left(t_{k}\right)}^{T}+Q$$
where Q represents the process noise matrix of the system, which is an adjustable parameter.

(II)Measurement Update

Step 1: Perform Cholesky decomposition for the predicted value of the error covariance $P_{t_{k}}^{⋆}$ at time $t_{k}$:



(19)
$$P_{t_{k}}^{⋆}={{S_{t_{k}}^{⋆}S}_{t_{k}}^{⋆}}^{T}$$



Step 2: Select cubature point:



(20)
$${\tilde{\chi }}_{i,t_{k}}=S_{t_{k}}^{⋆}\xi_{i}+\chi_{i,t_{k-1}}^{⋆}$$



Step 3: Propagate the cubature point through Equations (10)–(12):



(21)
$$Z_{i,t_{k}}^{⋆}=h({\tilde{\chi }}_{i,t_{k}})$$



Step 4: Estimate the predicted value of measurement at time $t_{k}$:



(22)
$$Z_{ship}^{⋆}\left(t_{k}\right)=\frac{1}{2n}\sum_{i=1}^{2n}Z_{i,t_{k}}^{⋆}$$



Step 5: Estimate the autocorrelation covariance matrix:

(23)$$P_{{zz|t}_{k}}=\frac{1}{2n}\sum_{i=1}^{2n}Z_{i,t_{k}}^{⋆}{Z_{i,t_{k}}^{⋆}}^{T}-Z_{ship}^{⋆}\left(t_{k}\right){Z_{ship}^{⋆}\left(t_{k}\right)}^{T}+R$$
where R represents the measurement noise matrix of the system, which is also an adjustable parameter.

Step 6: Estimate the cross-correlation covariance matrix:



(24)
$$P_{{xz|t}_{k}}=\frac{1}{2n}\sum_{i=1}^{2n}{\tilde{\chi }}_{i,t_{k}}{Z_{i,t_{k}}^{⋆}}^{T}-{Χ}_{ship}^{⋆}\left(t_{k}\right){Z_{ship}^{⋆}\left(t_{k}\right)}^{T}$$



Step 7: Estimate the Kalman gain:



(25)
$$W_{t_{k}}=P_{{xz|t}_{k}}P_{{zz|t}_{k}}^{-1}$$



Step 8: Estimate the state at time $t_{k}$:



(26)
$${\hat{Χ}}_{ship}\left(t_{k}\right)={Χ}_{ship}^{⋆}\left(t_{k}\right)+W_{t_{k}}(Z_{ship}\left(t_{k}\right)-Z_{ship}^{⋆}\left(t_{k}\right))$$



Step 9: Update the error covariance matrix at time $t_{k}$:



(27)
$$P_{t_{k}}=P_{t_{k}}^{⋆}-W_{t_{k}}P_{{zz|t}_{k}}W_{t_{k}}^{T}$$



#### 3.4.2. UKF Estimation Method

In contrast to the CKF estimation, the UKF estimation method depicts the Gaussian distribution of a nonlinear function via unscented transformation (UT). The algorithm flow is as follows $:P(t_{k-1})$.

Step 1: The UT is used to compute 2n+1 Sigma points $\zeta \left(t_{k-1}\right)$, the state weight matrix $\omega_{X}\left(t_{k-1}\right)$, and variance weight matrix $\omega_{P}\left(t_{k-1}\right)$:

(28)$$\zeta^{\left(1\right)}\left(t_{k-1}\right){=\hat{X}}_{ship}(t_{k-1})$$(29)$$\zeta^{\left(i\right)}\left(t_{k-1}\right){=\hat{X}}_{ship}\left(t_{k-1}\right)+\sqrt{\left(n+\lambda )P^{\left(i-1\right)}(t_{k-1}\right)}\;(i=2,3,\ldots ,n+1)$$(30)$$\zeta^{\left(i\right)}\left(t_{k-1}\right){=\hat{X}}_{ship}\left(t_{k-1}\right)-\sqrt{\left(n+\lambda )P^{\left(i-n-1\right)}(t_{k-1}\right)}\;(i=n+2,n+\;3,\ldots ,2n+1)$$(31)$${\omega_{X}}^{\left(1\right)}\left(t_{k-1}\right)=\frac{\lambda }{n+\lambda }$$(32)$${\omega_{P}}^{\left(1\right)}\left(t_{k-1}\right)=\frac{\lambda }{n+\lambda }+(1-\alpha^{2}+\beta )$$(33)$${\omega_{X}}^{\left(i\right)}\left(t_{k-1}\right)={\omega_{P}}^{\left(i\right)}\left(t_{k-1}\right)=\frac{\lambda }{2(n+\lambda )}\;(i=2,3,\ldots ,2n+1)$$
where $\cdot^{\left(i\right)}$ represents the *i*th column of the matrix. *n* = 12 represents the dimension of ${\hat{X}}_{ship}$. $\lambda =\alpha^{2}\left(n+\kappa \right)-n$ is the proportional parameter. α, β, and κ are adjustable parameters, and the values set in this paper are as follows: α=0.99999, β=0.3, and κ=3.

Step 2: One step predicts the state of Sigma points through Equation (9):



(34)
$$\zeta^{\left(i\right)}\left({t_{k}|t}_{k-1}\right)=f(\zeta^{\left(i\right)}\left(t_{k-1}\right))\;(i=1,2,3,\ldots ,2n+1)$$



Step 3: The Sigma points are weighted to obtain a one-step prediction of the state $X\left({t_{k}|t}_{k-1}\right)$ and variance matrix $P\left({t_{k}|t}_{k-1}\right)$:



(35)
$$X\left({t_{k}|t}_{k-1}\right)=\sum_{i=1}^{2n+1}\left[{\omega_{X}}^{\left(i\right)}\left(t_{k-1}\right)\zeta^{\left(i\right)}\left({t_{k}|t}_{k-1}\right)\right]$$


(36)
$$\begin{array}{ll} P\left(t_{k}|t_{k-1}\right)= & \sum_{i=1}^{2n+1}\{{\omega_{P}}^{\left(i\right)}\left(t_{k-1}\right)[X\left(t_{k}|t_{k-1}\right)\\ & -\zeta^{\left(i\right)}\left(t_{k}|t_{k-1}\right)]{\left[X\left({t_{k}|t}_{k-1}\right)-\zeta^{\left(i\right)}\left({t_{k}|t}_{k-1}\right)\right]}^{T}\}+Q \end{array}$$



Step 4: The first step is repeated to obtain a new set of Sigma points $\varsigma \left({t_{k}|t}_{k-1}\right)$ by performing UT on the one-step prediction of the state $X\left({t_{k}|t}_{k-1}\right)$ and variance matrix $P\left({t_{k}|t}_{k-1}\right)$.

Step 5: The observed value $\varrho \left({t_{k}|t}_{k-1}\right)$ of the Sigma point is obtained through Equations (10)–(12):



(37)
$$\varrho^{\left(i\right)}\left({t_{k}|t}_{k-1}\right)=h(\varsigma^{\left(i\right)}\left({t_{k}|t}_{k-1}\right))\left(i=1,2,3,\ldots ,2n+1\right)$$



Step 6: By weighting the observed values of the new Sigma point set, the mean value of the observation results, the auto-covariance matrix, and the cross-covariance matrix are obtained:



(38)
$$Z\left({t_{k}|t}_{k-1}\right)=\sum_{i=1}^{2n+1}\left[{\omega_{X}}^{\left(i\right)}\left(t_{k-1}\right)\varrho^{\left(i\right)}\left({t_{k}|t}_{k-1}\right)\right]$$


(39)
$$P_{\varrho ,\varrho }\left({t_{k}|t}_{k-1}\right)=\sum_{i=1}^{2n+1}\left\{{\omega_{P}}^{\left(i\right)}\left(t_{k-1}\right)[Z\left({t_{k}|t}_{k-1}\right)-\varrho^{\left(i\right)}\left({t_{k}|t}_{k-1}\right)]{\left[Z\left({t_{k}|t}_{k-1}\right)-\;\varrho^{\left(i\right)}\left({t_{k}|t}_{k-1}\right)\right]}^{T}\right\}+R$$


(40)
$$P_{\varsigma ,\varrho }\left({t_{k}|t}_{k-1}\right)=\sum_{i=1}^{2n+1}\left\{{\omega_{P}}^{\left(i\right)}\left(t_{k-1}\right)[Z\left({t_{k}|t}_{k-1}\right)-\varsigma^{\left(i\right)}\left({t_{k}|t}_{k-1}\right)]{\left[Z\left({t_{k}|t}_{k-1}\right)-\;\varrho^{\left(i\right)}\left({t_{k}|t}_{k-1}\right)\right]}^{T}\right\}$$



Step 7: The Kalman gain is estimated:



(41)
$$K\left(t_{k}\right)=P_{\varsigma ,\varrho }\left({t_{k}|t}_{k-1}\right){P_{\varrho ,\varrho }\left({t_{k}|t}_{k-1}\right)}^{T}$$



Step 8: The ship motion state ${\hat{X}}_{ship}\left(t_{k}\right)$ and covariance $P\left(t_{k}\right)$ are updated:



(42)
$${\hat{X}}_{ship}\left(t_{k}\right)=X\left({t_{k}|t}_{k-1}\right)+K\left(t_{k}\right)[Z_{ship}\left(t_{k}\right)-Z\left({t_{k}|t}_{k-1}\right)]$$


(43)
$$P\left(t_{k}\right)=P\left({t_{k}|t}_{k-1}\right)-K\left(t_{k}\right)P_{\varrho ,\varrho }\left({t_{k}|t}_{k-1}\right){K\left(t_{k}\right)}^{T}$$



### 3.5. Incremental LSTM Prediction Method

The LSTM neural network is a unique type of recurrent neural network (RNN), which may successfully combat the issue of gradient disappearance or explosion that typical RNN networks experience as training time and network layer rise. Therefore, the LSTM network is often used for timing prediction.

The LSTM network is often only used to forecast a ship’s track information in a ship track prediction application, and the projected step size is typically the equal time interval [25]. Its mapping connection may be expressed as follows:(44)$$\Omega \left(t+1\right)=f(\{\Omega \left(t-\kappa +1\right),\ldots ,\Omega \left(t-1\right),\Omega \left(t\right)\})$$
where κ represents the length of the historical sequence in the LSTM network.

Figure 5 depicts the network structure of the LSTM. Three gate structures make up the LSTM network. They are as follows:

(1) Forget gate: This is used to decide whether to keep or discard information. The information from the previous hidden state $h_{t-1}$ and the input information $\varepsilon_{t}$ are simultaneously processed via a sigmoid function to calculate the output value. The closer the result $f_{t}$ is to 0, the more it should be discarded. The forgetting factor can be calculated using the following formula:

(45)$$f_{t}=\sigma (W_{f}\cdot \left[h_{t-1},\varepsilon_{t}\right]+b_{f})$$
where $W_{f}$ is the weight matrix of the forgetting gate, $b_{f}$ is the offset term of the forgetting gate, and σ is the sigmoid function, which is specifically expressed as:(46)$$\sigma \left(net\right)=\frac{1}{1+e^{-net}}$$

(2) Input gate: This is used to update cell status. The information of the hidden state of the previous layer and the information of the current input are calculated via the sigmoid function to determine the type of updated information. Meanwhile, the information of the hidden state of the previous layer and the information of the current input are calculated via the *tanh* function to create a new candidate value vector. Finally, the output of the sigmoid function is multiplied by the output of the *tanh* function. The calculation formula is

(47)$$i_{t}=\sigma (W_{i}\cdot \left[h_{t-1},\varepsilon_{t}\right]+b_{i})$$(48)$${\overset{-}{C}}_{t}=tanh(W_{c}\cdot \left[h_{t-1},\varepsilon_{t}\right]+b_{c})$$
where $W_{i}$ and $W_{c}$ are the weight matrices of the input gate, $b_{i}$ and $b_{c}$ are the offset terms, and tanh represents the *tanh* function, which is specifically expressed as:(49)$$tanh\left(net\right)=\frac{e^{net}-e^{-net}}{e^{net}+e^{-net}}$$After passing through the forget gate and the input gate, the old cell state can be updated as
(50)$$C_{t}=f_{t}\cdot C_{t-1}+i_{t}\cdot {\overset{-}{C}}_{t}$$

(3) Output gate: This is used to determine the output value according to the cell state. First, the sigmoid function is used to determine the part of the cell state that needs to be output, then the cell state is calculated via the *tanh* function, and, finally, the output of network is multiplied. The calculation formula is

(51)$$o_{t}=\sigma (W_{o}\cdot \left[h_{t-1},\varepsilon_{t}\right]+b_{o})$$(52)$$h_{t}=o_{t}\cdot tanh(C_{t})$$
where $W_{o}$ is the weight matrix of the output gate, and $b_{o}$ is the offset term of the output gate. In Equations (47)–(52), parameters $W_{f}$, $W_{i}$, $W_{c}$, and $W_{o}$ and $b_{f},$
$b_{i},$
$b_{c},$ and $b_{o}$ are all parameters that can be adjusted.

Compared with the ship trajectory prediction method, the traditional NI-LSTM prediction relationship of the ship’s six degrees of freedom motion can be expressed as
(53)$$\eta_{ship}\left(t+1\right)=f(\{\eta_{ship}\left(t-\tau +1\right),\ldots ,\eta_{ship}\left(t-1\right),\eta_{ship}\left(t\right)\})$$

Some researchers utilize LSTM to predict ship motion attitude in addition to ship track prediction. Since ship attitude changes more irregularly and with a narrower change range than ship track, it is challenging to guarantee the accuracy of attitude prediction using solely LSTM. Researchers have made improvements to the LSTM network, including the addition of a self-attention mechanism for multi-scale prediction [26] and the decomposition of ship attitude data by superimposing the LSTM network with other prediction networks [27].

Although the aforementioned method can somewhat raise the LSTM network’s prediction accuracy, it requires many more calculations than a single LSTM network, which is not fast enough for the application in this study. On the other hand, the prediction accuracy of timing sequences with non-equal time intervals will be lower because LSTM is typically utilized for timing sequence prediction with equal time intervals. The time difference and the increment of the ship’s motion state is input into the network in this article using a single LSTM network structure. Considering that the speed of the ship also affects the change in motion state, the network also uses the ship’s speed $\nu_{ship}$ as an input. The mapping relationship can be expressed as follows:(54)$$\Gamma \left(t+1\right)-\Gamma \left(t\right)=f(\delta )$$
(55)$$\delta ={⋃}_{i=1}^{\tau }\left\{\Delta T_{i},\Delta \Gamma_{i},\nu_{ship}(t-i+1)\right\}$$
(56)$$\Delta \Gamma_{i}=\Gamma \left(t\right)-\Gamma (t-i+1)$$
(57)$$\Delta T_{i}=T_{predict}-T(t-i+1)$$
where $\Gamma =[\begin{array}{ccc} \hat{x} & \hat{y} & \begin{array}{ccc} \hat{\theta } & \hat{\varphi } & \hat{\psi } \end{array} \end{array}]$ is the latest estimate result ${\hat{Χ}}_{ship}$ of the CKF estimator for all elements except $\hat{z}$. $T_{predict}$ represents the moment to be predicted, and T(t−i+1) (*i* = 1, 2, 3,…,τ) represents the sequence of the latest τ estimated moments with the CKF estimator.

Since the observation of $z^{GPS}$ is set to zero in the observation Equation (10), its estimation with the CKF estimator is inaccurate. In order to avoid the problem of the prediction accuracy being reduced due to the inaccurate quantity, we did not take it as the input for the prediction network.

## 4. Results and Discussion

The experimental verification work is based on the ship model of the unmanned ship platform “CHCNAV APACHE 6”, which is equipped with a dual antenna GPS (installed at the fore and aft of the ship) for real-time measurement of the ship’s position information and an attitude instrument (installed close to the center of gravity of the ship) for real-time measurement of the ship’s attitude information. The experimental verification work was carried out in the Yanxi Lake experimental site, which is open water in a natural environment with an unknown random environment. Figure 6 displays the ship’s model and experimental site.

The navigation of an intelligent ship can be divided into two modes in the application: manual remote control and autonomous navigation. Different modes have different changing rules for the ship motion state. Furthermore, the characteristics of a ship’s motion state that change with speed vary. To verify the effectiveness and reliability of the NETIIP method, six sets of experiments were carried out at low speed, medium speed, and high speed, respectively, under the manual remote mode (MRM) and autonomous navigation mode (ANM).

Figure 7 shows the results of the real-time estimation of the ship’s position information (taking the north coordinates as an example) in the six trials by using the CKF estimator and UKF estimator. Herein, the longitude and latitude measured with the dual-antenna GPS are converted using the Universal Transverse Mercator (UTM) projection method. Meanwhile, the initial values of the error covariance ***P*** in Formulas (13), (29) and (30), the process noise matrix ***Q*** in Formulas (18) and (36), and the measurement noise matrix ***R*** in Formulas (23) and (39) are all set to the same value in order to compare the estimation accuracy of the two estimation techniques.

The CKF and UKF both have their own benefits and drawbacks for estimating ship positioning data in real-time. The initial convergence rate of the CKF is marginally quicker than the UKF’s for the same set of algorithm parameters (***a*** in Figure 7a–e). Nevertheless, the tracking effect of the CKF is worse than the UKF when a ship is turning with a high speed (***b*** in Figure 7c and ***a*** in Figure 7f).

Figure 8 shows the results of the real-time estimation of the ship’s attitude information (taking the pitch angle of ships as an example) in the six experiments by using the UKF estimator. Meanwhile, Figure 9 shows the CKF estimation results.

It can be seen that the UKF estimations have a low accuracy when the attitude angle is close to zero and fluctuates often (see Figure 8a). However, the estimated results can fundamentally converge towards the original data when the attitude angle is far from zero and changes relatively regularly. Comparatively, the CKF produces more accurate results than the UKF for the parameters given, and the CKF estimation method’s assessment of the ship’s attitude angle generally converges around the observed value. We are unable to determine which approach performs better in terms of estimation as the parameter settings may not be optimal for them. However, it is clear from a comparison of Figure 8 and Figure 9 that the UKF is more susceptible to changes in the features of the item to be estimated, making it more challenging to perform tasks such as parameter modification.

The results of the two approaches’ real-time estimations of the ship forward speed u are shown in Figure 10. The figure shows that while the UKF estimation results are nearly non-convergent, the CKF estimation results are smoother. Meanwhile, Equations (58) and (59) are used to calculate the resultant velocities and original observations to accurately evaluate the estimation accuracy for ship velocity with the two approaches, which is depicted in Figure 11.
(58)$$\nu_{estimation}\left(t_{k}\right)=\sqrt{{u\left(t_{k}\right)}^{2}+{v\left(t_{k}\right)}^{2}}$$
(59)$$\nu_{observation}\left(t_{k}\right)=\sqrt{{\left(\frac{x^{GPS}\left(t_{k+1}\right)-x^{GPS}\left(t_{k}\right)}{t_{k+1}-t_{k}}\right)}^{2}+{\left(\frac{y^{GPS}\left(t_{k+1}\right)-y^{GPS}\left(t_{k}\right)}{t_{k+1}-t_{k}}\right)}^{2}}$$

Equation (58) is the method for calculating the ship’s resultant velocity in the shipboard coordinate system, which is used for the vector addition of the estimated velocity. Meanwhile, Equation (59) is the method for calculating the resultant velocity in the geodetic coordinate system, which is appropriate for calculating the ship’s resultant velocity under the sensor’s original observation.

In contrast, Figure 10 and Figure 11 show that the results of the CKF estimator for the first-order state (ship speed) essentially converge towards the observations, while the results of the UKF estimator practically diverge.

The root-mean-square error (RMSE) coefficient (Equation (60)) is used to calculate the estimation results of each variable for assessment in order to more effectively evaluate the estimation accuracy of the estimated state variables using the UKF and CK.
(60)$$RMSE=\sqrt{\frac{1}{m}\sum_{i=1}^{m}{\left({\hat{\chi }}_{i}-\chi_{i}\right)}^{2}}$$
where m represents the number of estimated cycles, ${\hat{\chi }}_{i}$ represents the variable estimated for the *i*th period, and $\chi_{i}$ represents the truth value. However, because the true value of the quantity to be estimated cannot be acquired in the experiment, we accept the sensor’s observed value as the genuine value. The RMSE coefficient reflects the convergence and credibility of the estimated result. The smaller the value, the more the estimated result converges towards the observed value, and the higher its credibility. The RMSE coefficient of the estimated results is shown in Table 1.

Combined with Figure 7, Figure 8, Figure 9, Figure 10 and Figure 11 and the data in Table 1, under the same set of parameters, the UKF has a higher estimation accuracy than the CKF for ship position estimation. On the other hand, the UKF has better parameter robustness than the CKF. With the increase in ship speed, the estimation accuracy of the CKF decreases gradually; thus, for different ship speeds, the CKF must change various parameters to provide appropriate accuracy. However, the accuracy of the UKF estimation will not decrease with the increase in ship speed, that is, a set of parameters can adapt to the estimation of ship position at different speeds.

However, the CKF’s accuracy is substantially greater than the UKF’s for ship attitude. Even worse, the attitude estimated using the UKF does not steadily converge towards the observation (see Figure 8). Moreover, since attitude and ship velocity have a strong coupling connection, the UKF’s estimation for ship velocity virtually exhibits a diverging trend (see Figure 11). Thus, the UKF is not appropriate for the estimation of the multidimensional autocorrelation state.

Ship speed is one of the inputs for the prediction network in this study (Formula (55)), which has high requirements for the speed-related estimate accuracy of the pre-state estimation module. Therefore, the CKF is chosen as the pre-state estimate module rather than the UKF.

In order to verify the prediction accuracy of the NETIIP, the CKF estimations of the ship in automatic tracking mode with a six-knots set speed are first collected as the training set. The above six groups of data are used as the test set for prediction and compared with the prediction results of the NI-LSTM algorithm. The ship position prediction results are shown in Figure 12, and the attitude prediction results are shown in Figure 13. Therein, the prediction time is set to a random number with an upper limit of the estimation period of the CKF.

Comparing the prediction results of the two methods, it can be seen that the prediction result of NETIIP is closer to the truth data than NI-LSTM. On the other hand, with the change in ship speed, the accuracy of the NI-LSTM prediction also changes greatly. The prediction of the position (***a*** in Figure 12d) and attitude (Figure 13a,d) of the low-speed sailing mode almost diverges. In contrast, the NETIIP technique nearly converges towards the truth data and is less impacted by speed.

To better explore the degree of influence of the ship speed, the absolute value of the prediction error–velocity distribution is plotted, as shown in Figure 14 and Figure 15.

The prediction error of NI-LSTM tends to grow with the increase in ship speed in both the manual mode and automatic tracking mode, as can be seen from the distribution charts in Figure 14 and Figure 15, while the error of the NETIIP prediction method scarcely changes with ship speed. Since the training set chooses data from the automatic tracking mode, the prediction errors of the two approaches for predicting ship attitude are comparable for the test set in MRM, generally speaking, the NETIIIP algorithm is slightly superior to the NI-LSTM algorithm. In ANM, NETIIP maintains a high prediction accuracy, and its prediction error is essentially kept within 1° for the ship attitude. However, the prediction accuracy of NI-LSTM is poor, and its prediction error varies with the change in ship speed. The prediction results of the NI-LSTM algorithm almost exhibit a divergent trend throughout the range of 1–3 m/s, as observed when combined with Figure 13d and Figure 14b.

The RMSE coefficients under each mode are calculated in Table 2.

According to Equation (61) and the data in Table 2, the decrease ratios of the RMSE coefficient of the two prediction methods were calculated. The improving effect of the suggested prediction method on prediction accuracy is more noticeable the higher the value.
(61)$${RMSE}_{reduction}=\frac{{RMSE}_{NI-LSTM}-{RMSE}_{NETIIP}}{{RMSE}_{NETIIP}}\times 100\%$$

The decrease ratios of the RMSE coefficients of all the prediction state variables of each mode at the three speeds in Table 3 were averaged, as shown in Table 4, which represents the degree of improvement of the proposed prediction methods at different speeds.

The maximum ship velocities in the CKF estimator under each mode and those in the training set were obtained in accordance with Figure 11 in order to more accurately reflect the relationship between the degree of prediction accuracy improvement of NETIIP in order to more accurately reflect the relationship between the degree of prediction accuracy improvement and the change in ship speed, as shown in Table 5.

The accuracy of the prediction results of NETIIP is not significantly higher than the traditional NI-LSTM prediction method when the ship speed in the test set approaches that in the training set, as can be seen by comparing Table 4 and Table 5 and combining with the results of the prediction error distribution in Figure 14 and Figure 15. This is because the prediction accuracy of the traditional method is sufficient. However, the conventional NI-LSTM prediction technique has a low learning rate for the state change rule, and even fails to learn the proper rule, when the test set’s ship speed significantly differs from the training set’s. Therefore, the prediction error increases with the increase in the speed difference. Compared with the traditional NI-LSTM method, the prediction accuracy of the NETIIP prediction method is less affected by the speed. Therefore, compared with the traditional NI-LSTM prediction method, the improvement rate of the prediction accuracy increases with the increase in the speed difference.

It is vital to determine whether the time consumption of this technique is less than the need in order to confirm that the real-time performance of the suggested method satisfies the actual engineering requirements. According to Figure 1, the time of each cycle of the algorithm should not be longer than the sampling period of the sensor GPS, which is set to 0.1 s in this experiment. The NETIIP and NI-LSTM prediction methods are performed using software named MATLAB on a Windows 10 system with an Intel^®^ Core(TM) i5-3210m 2.50 Ghz processor. The average time consumption of the methods is calculated as shown in Table 4.

Because NETIIP uses a single LSTM network structure, which is the same as that of the NI-LSTM prediction technique, the algorithm time is comparable to that of NI-LSTM, and the average time is considerably less than the upper limit of the forecast time 0.1 s (see Table 6). Therefore, the real-time performance of the proposed prediction method meets the engineering needs.

## 5. Conclusions

This paper studied a non-equal time interval incremental prediction method for ship motion state to solve the problem of ship state estimation at different rates and sensor sampling times in intelligent ship navigation. First, as the method’s time state estimation modules, the estimation results of the CKF and UKF were first compared and studied, and the justifications for selecting the CKF estimator were given. Then, the prediction results of the NETIIP method and the traditional NI-LSTM prediction method were compared, and the impact of a change in ship velocity on the two prediction methods’ accuracy was examined. The comparative findings demonstrate that the suggested approach has good prediction accuracy for ship condition prediction under various modes and speeds when compared with the conventional NI-LSTM prediction methods. Meanwhile, the algorithm time is almost the same as that of the traditional prediction methods, and both can satisfy the requirements of actual engineering.

Nevertheless, the following are some drawbacks of the suggested approach in this paper: (a) In order to guarantee the algorithm’s prediction accuracy, the CKF estimation algorithm’s state estimation accuracy must first be verified. The CKF algorithm needs to adjust various sets of parameters for different speeds of the ships, which restricts how simple the prediction algorithm can be. (b) Different from the traditional NI-LSTM prediction method, the NETIIP algorithm needs to use time intervals as inputs for training and prediction, necessitating high-precision time stamps in the training set and test set. Inaccurate time stamps will lower the algorithm’s forecast accuracy.

As a result, future studies should take into account a more straightforward and efficient state estimation method as the prediction method’s equal time interval state estimation module. Furthermore, in order to lessen the effects of inadequate timestamp accuracy, a novel predictive compensating approach must be created.

## Figures and Tables

**Figure 1 sensors-23-02852-f001:**
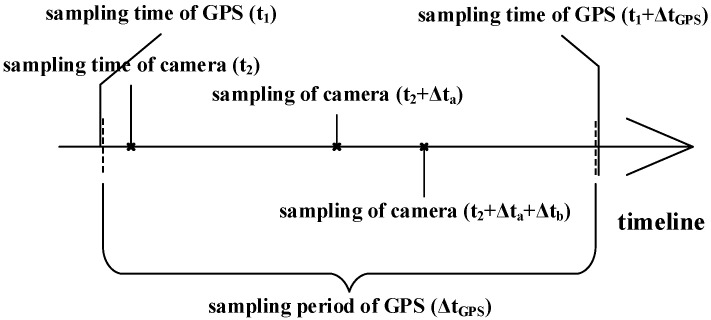
Multi-sensor system sampling diagram for MASS.

**Figure 2 sensors-23-02852-f002:**
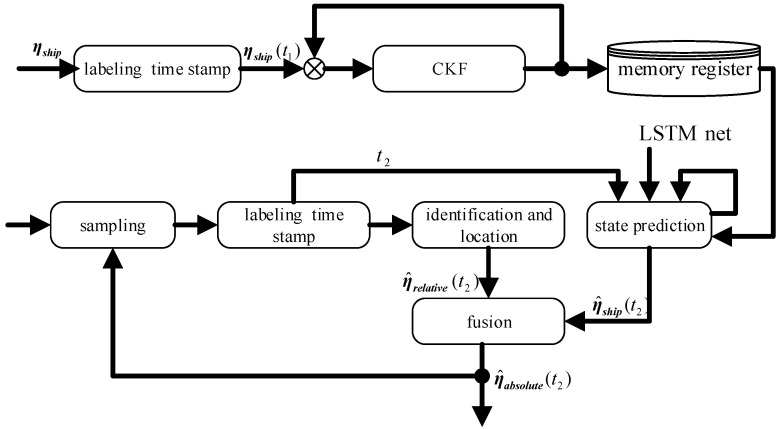
The algorithm flowchart of NETIIP.

**Figure 3 sensors-23-02852-f003:**
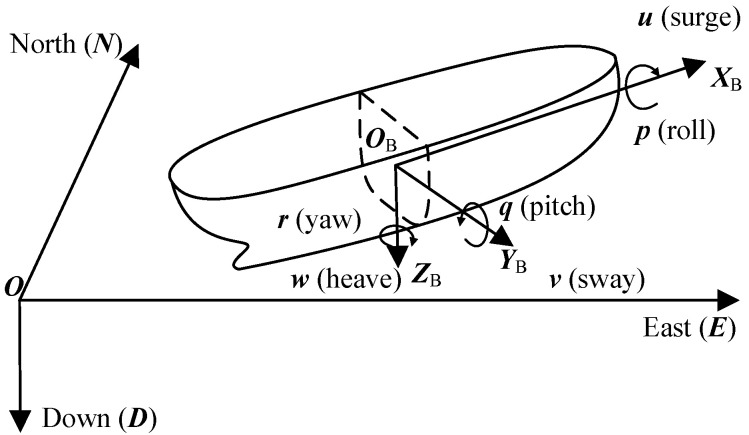
Coordinate system diagram.

**Figure 4 sensors-23-02852-f004:**
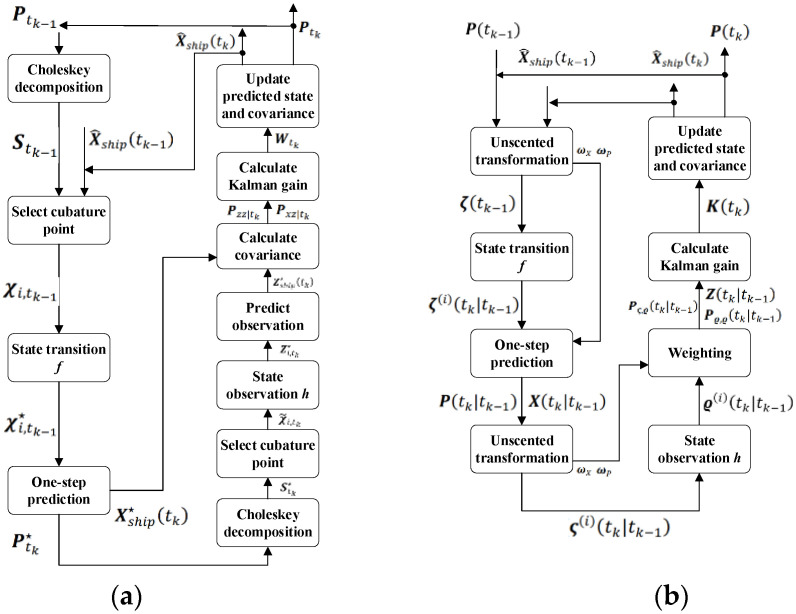
Algorithm flow charts: (**a**) CKF and (**b**) UKF.

**Figure 5 sensors-23-02852-f005:**
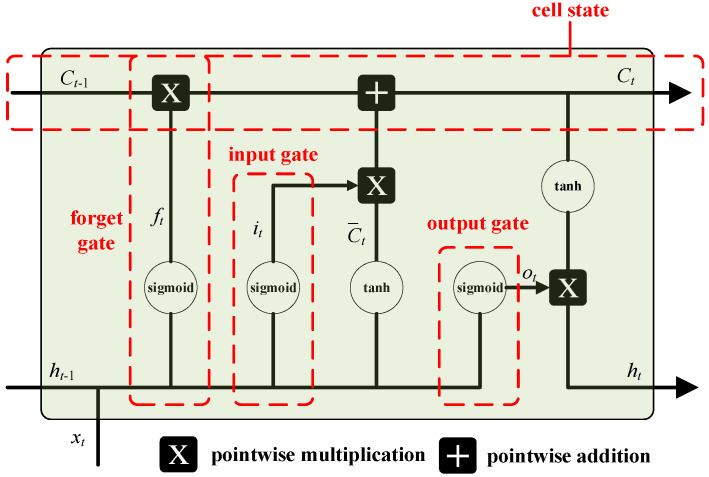
LSTM network structure.

**Figure 6 sensors-23-02852-f006:**
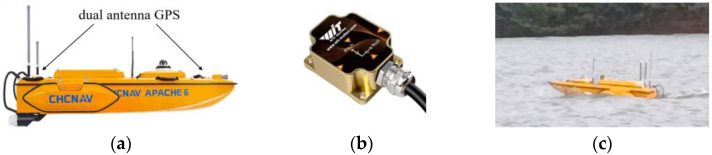
Experimental ship models and sensors: (**a**) experimental model, (**b**) attitude instrument, and (**c**) experimental site.

**Figure 7 sensors-23-02852-f007:**
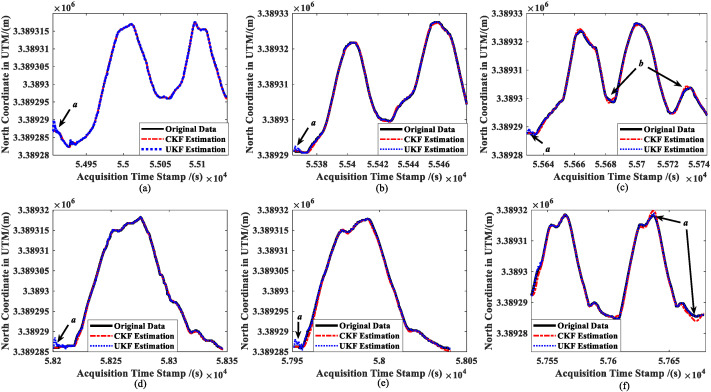
Comparison of ship position estimations: (**a**) slow speed of MRM, (**b**) medium speed of MRM, (**c**) high speed of MRM, (**d**) slow speed of ANM, (**e**) medium speed of ANM, and (**f**) high speed of ANM.

**Figure 8 sensors-23-02852-f008:**
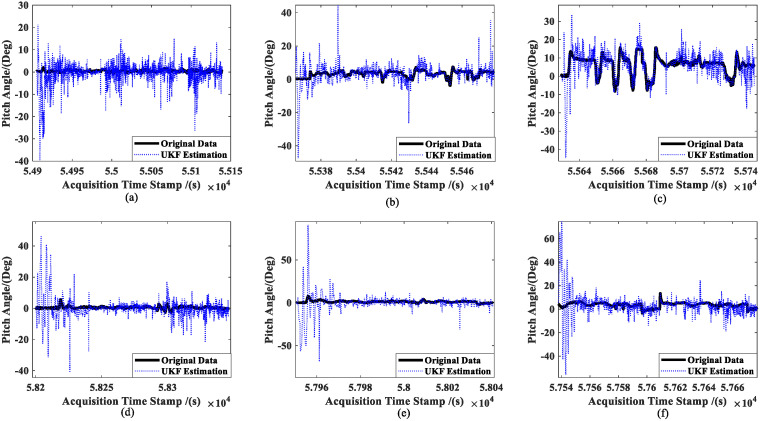
UKF estimation results for the ship’s attitude: (**a**) slow speed of MRM, (**b**) medium speed of MRM, (**c**) high speed of MRM, (**d**) slow speed of ANM, (**e**) medium speed of ANM, and (**f**) high speed of ANM.

**Figure 9 sensors-23-02852-f009:**
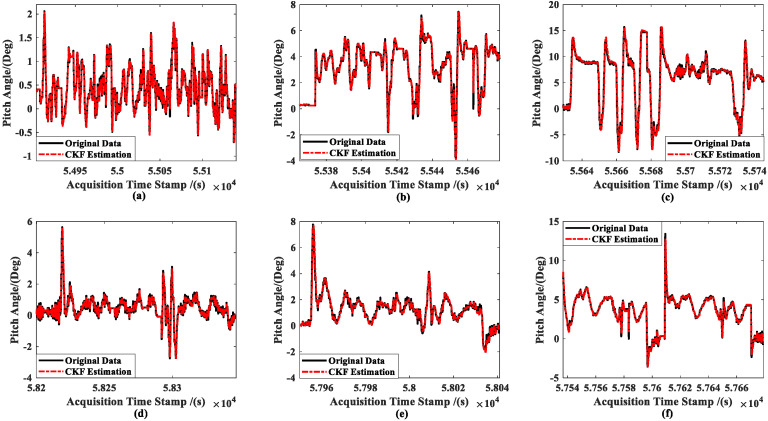
CKF estimation results for the ship’s attitude: (**a**) slow speed of MRM, (**b**) medium speed of MRM, (**c**) high speed of MRM, (**d**) slow speed of ANM, (**e**) medium speed of ANM, and (**f**) high speed of ANM.

**Figure 10 sensors-23-02852-f010:**
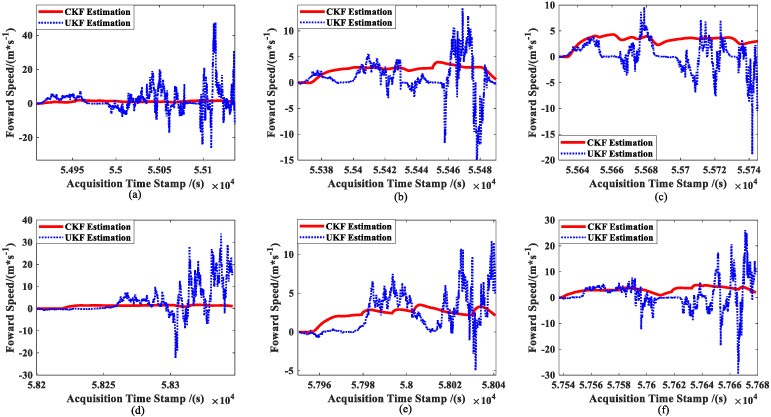
Comparison of ship forward speed estimations: (**a**) slow speed of MRM, (**b**) medium speed of MRM, (**c**) high speed of MRM, (**d**) slow speed of ANM, (**e**) medium speed of ANM, and (**f**) high speed of ANM.

**Figure 11 sensors-23-02852-f011:**
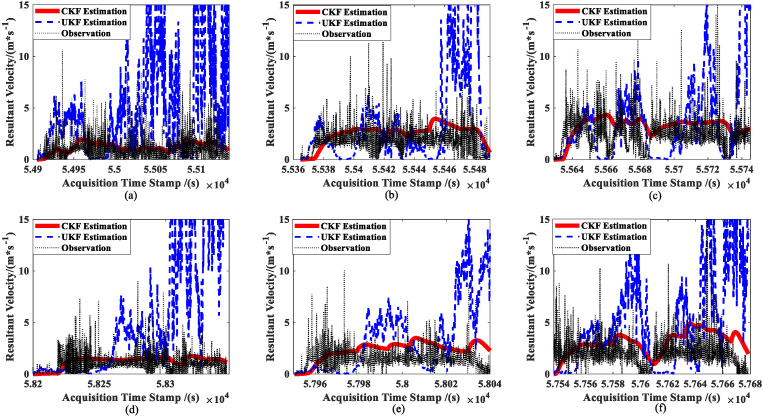
Comparison of ship resultant velocity estimations: (**a**) slow speed of MRM, (**b**) medium speed of MRM, (**c**) high speed of MRM, (**d**) slow speed of ANM, (**e**) medium speed of ANM, and (**f**) high speed of ANM.

**Figure 12 sensors-23-02852-f012:**
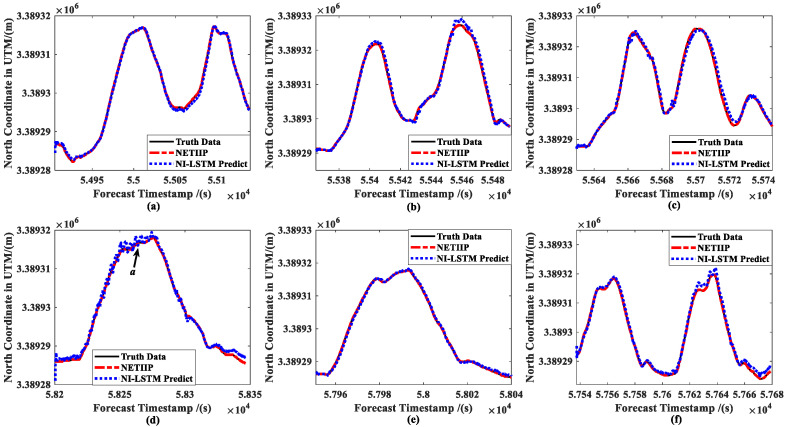
Comparison of ship position predictions: (**a**) slow speed of MRM, (**b**) medium speed of MRM, (**c**) high speed of MRM, (**d**) slow speed of ANM, (**e**) medium speed of ANM, and (**f**) high speed of ANM.

**Figure 13 sensors-23-02852-f013:**
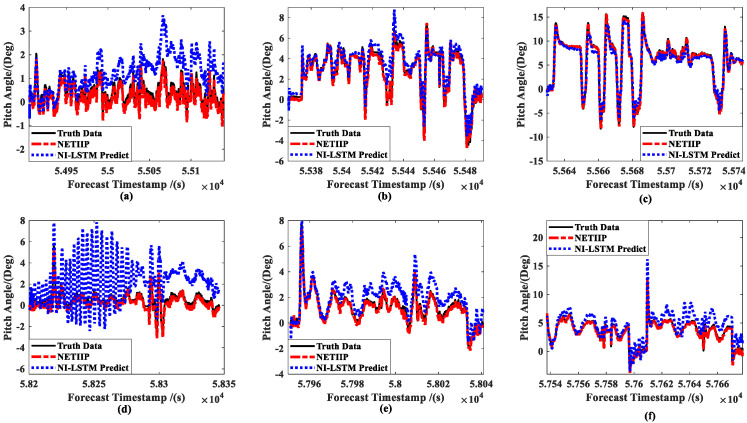
Comparison of ship attitude predictions: (**a**) slow speed of MRM, (**b**) medium speed of MRM, (**c**) high speed of MRM, (**d**) slow speed of ANM, (**e**) medium speed of ANM, and (**f**) high speed of ANM.

**Figure 14 sensors-23-02852-f014:**
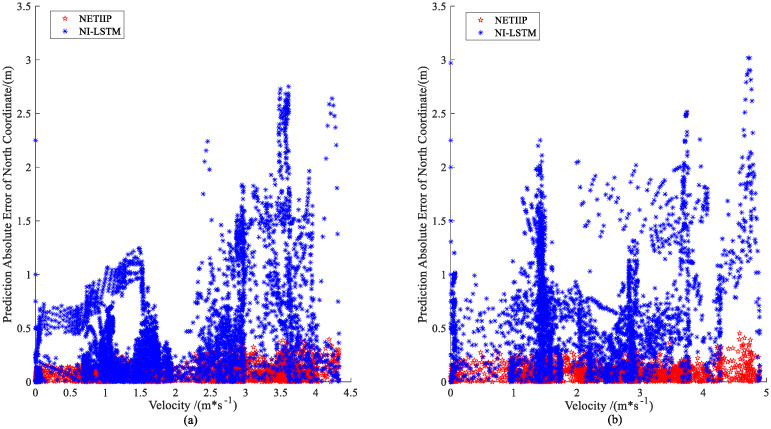
Prediction error–velocity distribution of ship position: (**a**) MRM and (**b**) ANM.

**Figure 15 sensors-23-02852-f015:**
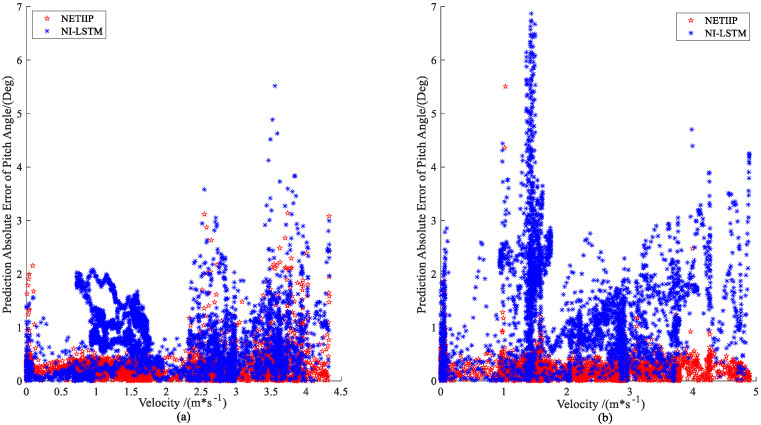
Prediction error–velocity distribution of ship attitude: (**a**) MRM and (**b**) ANM.

**Table 1 sensors-23-02852-t001:** RMSE coefficients of different models and different estimators.

Estimator		North	East	Pitch (Rad)	Roll (Rad)	Head (Rad)
CKF	MRM_Slow	0.2081	0.1614	0.0013	0.0021	0.1911
	MRM_Medium	0.4058	0.4747	0.0053	0.0059	0.4508
	MRM_Fast	0.5097	0.7541	0.0110	0.0093	0.3328
	ANM_Slow	0.2378	0.2105	0.0031	0.0097	0.3885
	ANM_Medium	0.4869	0.4133	0.0035	0.0104	0.5389
	ANM_Fast	0.8318	0.5225	0.0059	0.0087	0.4275
UKF	MRM_Slow	0.2278	0.0883	4.0196	2.6769	0.6209
	MRM_Medium	0.1466	0.1160	2.3857	2.7318	1.1448
	MRM_Fast	0.1690	0.1322	1.8902	3.2087	0.9575
	ANM_Slow	0.1005	0.2516	4.1937	3.3752	1.5118
	ANM_Medium	0.2384	0.2239	3.8999	3.5797	1.4721
	ANM _Fast	0.1494	0.1753	2.9908	3.0315	1.0680

**Table 2 sensors-23-02852-t002:** RMSE coefficients of different models and different predictions.

		North	East	Pitch (Rad)	Roll (Rad)	Head (Rad)
NETIIP	MRM_Slow	0.0823	0.0489	0.0039	0.0030	0.2336
	MRM_Medium	0.1079	0.0973	0.0055	0.0054	0.2599
	MRM_Fast	0.1333	0.1349	0.0104	0.0063	0.1927
	ANM_Slow	0.0835	0.0552	0.0050	0.0070	0.1741
	ANM_Medium	0.0994	0.0938	0.0046	0.0070	0.2214
	ANM_Fast	0.1197	0.1132	0.0060	0.0056	0.1753
NI-LSTM	MRM_Slow	0.4396	0.3611	0.0167	0.0193	0.8163
	MRM_Medium	0.7770	0.7772	0.0130	0.0190	1.0182
	MRM_Fast	1.0562	2.4098	0.0187	0.0257	1.6340
	ANM_Slow	0.8328	1.4724	0.0440	0.0218	1.3956
	ANM_Medium	0.3952	0.7633	0.0162	0.0138	0.5420
	ANM_Fast	1.0255	0.9913	0.0253	0.0177	0.7663

**Table 3 sensors-23-02852-t003:** Decrease ratio of RMSE coefficient of each mode (%).

	North	East	Pitch	Roll	Head
MRM_Slow	81.28	86.50	76.65	84.45	71.38
MRM_Medium	86.10	87.48	57.70	71.58	74.47
MRM_Fast	87.38	94.40	44.38	75.50	88.21
ANM_Slow	89.97	96.25	88.63	67.89	87.53
ANM_Medium	74.80	87.71	71.60	49.28	59.15
ANM_Fast	88.33	88.60	76.30	68.36	77.12

**Table 4 sensors-23-02852-t004:** Average decrease ratio of RMSE coefficient (%).

**Speed Mode**	Slow	Medium	Fast	**Average**
**Ratio (%)**	83.25	71.99	78.858	78.03

**Table 5 sensors-23-02852-t005:** Maximum velocities for each mode (m/s).

MRM			ANM			Training Set
Slow	Medium	Fast	Slow	Medium	Fast	——
1.9383	3.9522	4.3282	1.7456	3.4997	4.8914	3.2647

**Table 6 sensors-23-02852-t006:** Average time consumption.

Predictor	MRM			ANM		
	Slow	Medium	Fast	Slow	Medium	Fast
NETIIP	0.02297	0.02318	0.0218	0.0244	0.0215	0.0234
NI-LSTM	0.0223	0.0225	0.0229	0.02338	0.0227	0.0227

## Data Availability

Not applicable.

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
