# Peer review of "A Non-Equal Time Interval Incremental Motion Prediction Method for Maritime Autonomous Surface Ships"

_sensors, 2023, doi:10.3390/s23052852_

Round 1

Reviewer 1 Report

This paper proposed a non-equal time interval incremental prediction method of ship motion state to solve the problem of ship state estimation at different sampling rate sensor sampling times in ship intelligent navigation.The cubature Kalman filter(CKF) is employed to estimate the ship's motion at equal intervals,anda long short-term memory network(LSTM) is used to predict the ship's motion state.However, there are still a few minor problems in the manuscript.

(1) In "1 Introduction ", which is relatively long for this article. The author can briefly introduce the relevant background.

(2) There is a symbol error in formula (1) on line 235.

(3) Whether EKF(Extended Kalman Filter) can also work here, and whether the adopted CKF (Unscented Kalman Filter) is superior to the EKF?

(4) In the simulation experiment, the authors onlytake the pitch angle of ships as an example.How about sway, roll, and other degrees of freedom?

Author Response

Dear editors and reviewers,

Thank you for your letter and the reviewers’ comments on our manuscript entitled " A Non-equal Time-interval Incremental Motion Prediction Method for Maritime Autonomous Surface Ships " (Manuscript ID: sensors-2227379). Those comments are very helpful for revising and improving our paper, as well as the important guiding significance to other research. We have studied the comments carefully and made corrections which we hope meet with approval. The responds to the reviewers’ comments are as follows (the replies are highlighted in blue).

Replies to the reviewers’ comments:

Reviewer #1:

We would like to thank you very much for your detailed comments and suggestions. The following is our detailed reply.

Comment 1: In “1 Introduction”, which is relatively long for this article. The author can briefly introduce the relevant background.

Response: Thanks for this comment. The first part of the previous manuscript, titled "1 Introduction" has been slightly modified. While keeping a brief introduction of the background and the advantages of the previous work of other researchers in this section, we have taken out the related work that is pertinent to the paper's main topic and placed it in the second part, titled "2 Related Works" (See the "Revised Manuscript " for details). This section analyses in depth the benefits and drawbacks of the pertinent work as well as its relationship to our research.

Comment 2: There is a symbol error in formula (1) on line 235......

Response: Thanks for this criticism. However, after carefully reviewing the formulation of formula (1) in the previous manuscript, we were able to conclude that there was no error:

Thus, we think that the formula might not be shown appropriately as a result of the different document conversion or opening style. We will remind the journal editor about this. Once again, I appreciate your correction.

Comment 3: Whether EKF (Extended Kalman Filter) can also work here, and whether the adopted CKF is superior to the EKF.

Response: Thanks for this comment. Because of the following factors, we decided not to include the EKF section in the updated manuscript.

On the one hand, numerous academics have previously compared and confirmed the EKF and UKF approaches from different perspectives, including theory and comparative experiment, and have come to the conclusion that UKF has higher estimation accuracy than EKF and is also simpler to implement [1, 2]. Therefore, in this paper, we use UKF method as a comparison method, which can further demonstrate the applicability and reliability of CKF method. In Part 3.4 of the revised paper, we add the following statement: “As nonlinear filtering methods, both EKF and UKF can realize real-time state estimation of nonlinear ship motion system. While UKF has the advantages of faster convergence, higher filtering precision and easier realization [23]”.

On the other hand, although EKF can work in nonlinear systems, it is not suitable for strongly nonlinear systems. For the state-space equation established in the manuscript (formula (9)), it can be seen that if EKF method is used for estimation, partial derivative of the coefficient of the state vector  in this equation with respect to  is needed to calculate the Jacobian matrix. According to formula (3) - (8), it can be seen that the coefficient has nonlinear relationship with some variables of the state vector , such as , , , but not all variables in the vector. So the partial derivative is difficult to solve and not easy to achieve.

We think it will be challenging to verify the addition of the EKF algorithm to the experiment for the two reasons mentioned above. When compared to the CKF method, the UKF algorithm is more convincing.

Reference:

[1] Chen, Z; Bai, M; Lei, J; Huang, Y; Wang, J; Xia, X. Comparison of UKF and EKF Filter Algorithm in INS/BDS Tightly Mode. 30th Chinese Control and Decision Conference, Shenyang, China, 9th June 2018.

[2] Song, L; Ji, H. Application and comparison of UKF and EKF algorithm in target tracking with multiple passive sensors. Systems Engineering and Electronics 2009, 31, 1083-1086.

Comment 4: In the simulation experiment, the authors only take the pitch angle of ships as an example. How about sway, roll, and other degrees of freedom?

Response: Thanks for this comment. First of all, we are sorry for missing Figure 11 in the previous manuscript. Due to the conversion of manuscript format, Figure 11 was not put into the manuscript, which is also our mistake in the manuscript proofreading, for which we are deeply sorry.

On the other hand, the results we put in the section “4. Results and Discussion” are representative. In formula (3) - (4) of the paper, the 6-degree of freedom state is divided into two parts,  and . Therefore, we show the prediction results of partial results (North Coordinate  - in , as shown in Figure 11, pitch angle  in , as shown in Figure 12) in the “4. Results and Discussion”.

We evaluated the correlation coefficient (such as the RMSE coefficient) of each degree of freedom for the prediction results of other degrees of freedom to assess the advantages and disadvantages of the prediction results (see Table 2-3). By doing this, it is possible to demonstrate that the suggested prediction approach has a higher prediction accuracy than the conventional method for states with different degrees of freedom while simultaneously guaranteeing the paper's attractive layout. In the meantime, we included the comparison graph of the six degrees of freedom prediction findings in the attached document named “Model Experiment Prediction Result Figures” for the experts' reference to demonstrate the validity of the data.

Reviewer 2 Report

The paper deals with motion prediction of the maritime autonomous ship. Paper is interesting and deals with actual problems.

There are mentions about GPS, GNSS and BeiDou. I suppose that systems like GPS, Glonass and BeiDou are subsets of GNSS.

Paper describes problems of data fusion (GPS, Lidar, etc.), but this is not explained further in the text...

Also paper describes various sampling frequency in various sensors. Is this a problem in controlling the ship? I suppose movement of the ship is very slow according to the sampling frequency e.g. GPS that is 5Hz or camera that is 20Hz.

Instead of "median speed" I think there should me "medium speed"

Reviewer 3 Report

Dear Colleagues,

Would you please consider the following comments:

1- The paper needs a proofreading for English grammar and some expressions. Special attention should be given to sentences yellow highlighted.

2- It would be useful for readers to get idea about the flowchart for both CKF and UKF

3- On page 11, line 475, and page 15, line 563: the acronym (METIIP) should be corrected to NETIIP

4- On page 11, figure 6: the results obtained using UKF seems good enough compared to those results obtained using CKF. 

5- on page 12, line 486-490: the argument given in this paragraph with regard to CKF and UKF is not so clear. Why CKF converges faster than UKF, and why UKF has better universality?

6- On page 13, from figure 9: why UKF shows unstable estimation? 

7- On page 13, from figures 9 and 10: would it be possible to show the real data (true data)

8- On page 14, line 520 (highlighted sentence): not clear

9- On page 14, equations (42) and (43) : what is the difference between equation (42) and equation (43)? 

10- On page 15, figure 12: it seems that the obtained results with manual remote mode (MRM) for both NETIIP and NI-LSTM are coincided and there is an acceptable match, would you please provide causes for this behavior.

11- On page 15, figure 12: for Auto navigation Mode (ANM), there seems to be a constant offset between the results of NETIIP and NI-LSTM, (why)?

12- On pages 15 and 16: from figures 13 and 14: the prediction error of Pitch angle seems positive only, does it the actual case?, is there negative value?

13- On page 16, from figure 14: the results of the prediction pitch error for the case of ANM seems much better for the case of MRM, but from figure 12 on page 15, the case is opposite? (any reason for this behavior)?

Best Regards
